# Human Antibody Domains and Fragments Targeting Neutrophil Elastase as Candidate Therapeutics for Cancer and Inflammation-Related Diseases

**DOI:** 10.3390/ijms222011136

**Published:** 2021-10-15

**Authors:** Xiaojie Chu, Zehua Sun, Du-San Baek, Wei Li, John W. Mellors, Steven D. Shapiro, Dimiter S. Dimitrov

**Affiliations:** 1Center for Antibody Therapeutics, Division of Infectious Diseases, Department of Medicine, University of Pittsburgh School of Medicine, Pittsburgh, PA 15261, USA; ZES20@pitt.edu (Z.S.); DUB5@pitt.edu (D.-S.B.); LIWEI171@pitt.edu (W.L.); jwm1@pitt.edu (J.W.M.); 2Abound Bio, Pittsburgh, PA 15219, USA; 3Department of Medicine, University of Pittsburgh School of Medicine, Pittsburgh, PA 15261, USA; shapirosd@upmc.edu; 4University of Pittsburgh Medical Center (UPMC), Pittsburgh, PA 15261, USA

**Keywords:** therapeutic antibodies, neutrophil elastase, inflammatory disease, cancer

## Abstract

Neutrophil elastase (NE) is a serine protease released during neutrophil maturation. High levels of NE are related to lung tissue damage and poor prognosis in cancer; thus, NE is a potential target for therapeutic immunotherapy for multiple lung diseases and cancers. Here, we isolate and characterize two high-affinity, specific, and noncompetitive anti-NE antibodies Fab 1C10 and V_H_ 1D1.43 from two large phage-displayed human Fab and V_H_ libraries. After fusion with human IgG1 Fc, both of them (V_H_-Fc 1D1.43 and IgG1 1C10) inhibit NE enzymatic activity with V_H_-Fc 1D1.43 showing comparable inhibitory effects to that of the small molecule NE inhibitor SPCK and IgG1 1C10 exhibiting even higher (2.6-fold) activity than SPCK. Their epitopes, as mapped by peptide arrays combined with structural modeling, indicate different mechanisms for blocking NE activity. Both V_H_-Fc and IgG1 antibodies block NE uptake by cancer cells and fibroblast differentiation. V_H_-Fc 1D1.43 and IgG1 1C10 are promising for the antibody-based immunotherapy of cancer and inflammatory diseases.

## 1. Introduction

Neutrophil elastase (NE) is a serine protease released by neutrophil degranulation or during the formation of a neutrophil extracellular trap (NET) [1,2]; it is generally considered the main contributor of neutrophil protease activity. Neutrophils are the most abundant white blood cells and play a major role in host defense against bacterial infection [3]. They can rapidly release cytokines, chemokines, reactive oxygen species, and proteases to help defend against bacterial infection and regulate inflammation [4]. However, the prolonged activation of neutrophils contributes to the pathophysiological changes in lung and causes acute or chronic inflammatory diseases such as chronic obstructive pulmonary disease (COPD) [5], cystic fibrosis [6], acute lung injury [7], and acute respiratory distress syndrome bronchiectasis [8]. NE, as an important regulator of the inflammatory response [9], can degrade all of the extracellular matrix proteins such as elastin, collagens, fibronectin, and lung surfactant protein and can also activate lung epithelial cells to produce inflammatory cytokines which could further activate neutrophils, then cause acute lung injury or fibrosis [10,11]. The level of NE activity and the cell count of neutrophils were highly elevated in sputum plasma from cystic fibrosis patients compared with healthy individuals [12,13]. Similarly, COPD patients also have a two-fold higher level of NE activity than healthy individuals in serum and bronchoalveolar lavage (BAL) fluid [14]. Despite respiratory diseases, many studies have shown that NE could also promote tumor proliferation through degrading IRS-1 and activating the PI3K–AKT signal pathway [15,16]. High NE activity is an indicator of poor prognosis in breast cancer, colorectal cancer, and non-small cell lung cancer [17,18,19,20]. In addition, NE activity could promote the adhesion of pancreatic cancer cells to vascular endothelial cells through stimulating the E-selectin expression [21]. Thus, the inhibition of neutrophil elastase activity can be considered a novel druggable strategy in cancer and inflammatory diseases.

Several NE inhibitors have been evaluated in mouse cancer models, including sivelestat and curcumin. Even though NE inhibitors inhibit NE activity in vitro characterized on cancer cells such as gastric cancer and breast cancer [22,23], its function on reducing cancer growth is still minimal [24,25]. Additionally, in inflammation-related diseases, although the sivelestat could decrease the incidence of acute lung injury after surgery [26] and the AZD 9668, another NE inhibitor, which has been proven to be able to improve lung function with bronchiectasis and reduce inflammatory markers in sputum for COPD patients [27], the benefit in acute lung injury and acute respiratory distress syndrome is still limited. In recent years, many novel orally available neutrophil elastase inhibitors have been designed and some have already entered phase II clinical trials for a variety of pulmonary diseases. MPH-966 significantly suppressed NE activity and reduced pro-inflammatory cytokines in a 5-FU-induced intestinal mucositis mouse model [28]. In a preclinical study, BAY-85-8501 proved to be able to prevent the development of lung injury and inflammation induced by NE in acute lung injury [29]. A phase I clinical study also showed a safety and tolerability profile in patients with non-cystic fibrosis bronchiectasis [30]. POL6014, a novel NE inhibitor developed for patients with cystic fibrosis, showed a safety and tolerability profile and a significant reduction in active NE after single dosing in a phase I clinical trial [31]. However, all these studies were designed for pulmonary diseases. We are not aware of any cancer-related studies.

In many cancer types, both the expression and activity of neutrophil elastase are upregulated. The number of neutrophils and the neutrophil to lymphocyte (N/L) ratio in peripheral blood is significantly higher in patients with lung cancer with or without COPD than in patients with COPD or healthy individuals. However, the cell count of neutrophils in BAL fluid is significantly lower in patients with lung cancer than in individuals with COPD. Notably, the NE level in serum and BAL fluid is five and threefold, respectively, greater in lung cancer patients compared to patients with COPD [14]. A strong NE proteolytic fingerprint was also found in the colon adenocarcinoma proteome [32], and elevated NE has a positive correlation with the poor response to trastuzumab therapy [33]. In summary, cancer is a potent inducer of NE. Nowadays, immunotherapy is considered to be a promising therapy in addition to traditional therapies, including chemotherapy, surgery, and radiotherapy. Antibody-based immunotherapy is attractive for cancer therapy for its high specificity, its high-affinity to the targets at nM levels, and the ability to activate effector functions elicited by the Fc region. In recent decades, there has been a growing interest in pursuing antibody fragments and domains as therapeutics, including antigen-binding fragments (Fab, 50 kDa), single-chain variable fragments (scFv, 30 kDa), and V_H_ (heavy-chain variable domain, 15 kDa) by the virtual of their smaller molecular sizes, and desirable pharmacokinetics in special clinical applications [34,35]. In addition, their low immunogenicity, high stability, and small size make the domain antibody much easier to infiltrate into cancer tissue and block the antigen inside the tumor.

The mechanism by which antibody-neutralizing enzymes may much differ from enzymatic inhibitors depends on the antibody-binding epitope. Firstly, same as enzyme inhibitors, an antibody can directly block the enzyme active site to inhibit enzyme activity [36,37]. Secondly, an antibody can interact with the adjacent regions of the active site to obstruct the access of substrates [38] or interact with other regions which could induce a conformational change and, thus, deactivating its catalytic activity [37]. In our current study, we identify potent NE monoclonal antibodies by panning our large human V_H_ and Fab antibody phage libraries against recombinant neutrophil elastase. Several binders are screened and characterized for affinity and stabilities. Among these, Fab 1C10 and V_H_ 1D1.43 with the IgG1 Fc fusion format show specificity against NE and have potent inhibition effects on enzyme activity. To our best knowledge, this is the first report of NE-specific antibodies as a new format of neutrophil elastase inhibitors with potential implications for immunotherapy on cancer and inflammatory diseases.

## 2. Results

### 2.1. Selection of High-Affinity V_H_ Domains and Fab Antibodies against Neutrophil Elastase

We, previously, developed several large phage-displayed libraries using PBMC from non-immunized healthy donors [39,40]. In this study, two large phage-displayed human V_H_ and Fab libraries were used for panning against recombinant NE protein, which fuses with the human IgG1 Fc tag due to the enhanced stability. After three rounds of panning, a panel of VH and Fab binders was screened. Among these binders, two high-affinity antibodies, termed as Fab 1C10 and V_H_ 1D1, were selected based on their high affinities and favorable biophysical properties. The EC_50_ of Fab 1C10 and V_H_ 1D1 tested by ELISA were 16.3 ± 0.52 nM and 25.5 ± 2.57 nM, respectively (Figure 1a). The equilibrium dissociation constant (KD) of these two binders (Fab 1C10 and V_H_ 1D1) with recombinant human ELA2 measured by BLItz was 8.5 nM and 69.7 nM, respectively (Table 1). To further improve the affinity of the V_H_ 1D1 binder, we constructed a random mutant library by using error-prone PCR. After three rounds of panning against NE-Fc with kinetic selection by using the prototype V_H_ 1D1, one clone named V_H_ 1D1.43 was identified. The EC_50_ of V_H_ 1D1.43 tested by ELISA was 4.3 ± 0.17 nM (Figure 1a) and the equilibrium dissociation constant was 8.4 nM, 5-10-fold higher than that of the parental clone (Table 1). In addition, we evaluated the aggregation propensity of these binders by using dynamic light scattering (DLS) and size-exclusion chromatography (SEC). The V_H_ 1D1 had both monomer and dimer forms, but after affinity maturation, the main form of V_H_ 1D1.43 was a monomer measured by SEC (Figure 1b) and there was no obvious aggregation after a 7-day incubation at 37 °C as measured by DLS (Figure 1d). Fab 1C10 was a monomer and had no high molecular weight form measured by SEC (Figure 1c). Even though Fab 1C10 showed a high molecular peak by DLS at 4 °C, after the 7-day incubation at 37 °C at a concentration of 1 mg/mL, there was no aggregation observed by DLS, indicating that the high molecular form was reversible and soluble (Figure 1d).

### 2.2. Conversion of V_H_ Domain and Fab Binders to V_H_-Fc and IgG1

To increase the avidity and extend the in vivo half-life of V_H_ 1D1.43 and Fab 1C10, these two binders were converted to a bivalent antibody by fusion to the human IgG1 Fc (V_H_-Fc 1D1.43 and IgG1 1C10). The EC_50_ of IgG1 1C10 and V_H_-Fc 1D1.43 tested by ELISA were 0.4 ± 0.02 nM and 0.66 ± 0.03 nM, respectively (Figure 1a). The equilibrium dissociation constant (KD) of V_H_-Fc 1D1.43 and IgG1 1C10 measured by BLItz was 1.9 nM and 0.14 nM, respectively (Figure 1e,f, Table 1). The forms of IgG 1C10 and V_H_-Fc 1D1.43 were a monomer and had no high molecular weight species measured by SEC (Figure 1b,c).

### 2.3. Specificity of V_H_ Domain and Fab Binders with Human NE

To further test the specificity of these two V_H_ domain and Fab binders, we tested the binding effects with recombinant human myeloperoxidase protein (rhMPO) and Proteinase 3 (PR3), both of which were released by neutrophils. The ELISA results showed that Fab 1C10 and V_H_ 1D1.43 specifically bound to NE and did not bind to other enzymes released by neutrophils (Figure 2a,b). We further tested the specificity of IgG1 1C10 and V_H_-Fc 1D1.43 by the detection of binding to BSA; the results showed that the binders had no non-specific binding with other proteins (Figure 2c,d).

### 2.4. Inhibition on NE Enzyme Activity and Function

To study the function of the binders, we tested the inhibition effect on neutrophil elastase enzyme activity. The estimated IC_50_ of V_H_ 1D1.43 was about 60 uM, but after converting to the V_H_-Fc form, the IC_50_ was 5.3 ± 1.2 uM, a 10-fold elevation compared to the V_H_ form (Figure 3a,c). Additionally, the estimated IC_50_ of Fab 1C10 was about 5 uM, and after converting to IgG1, the IC_50_ was elevated to 1.8 ± 0.9 uM (Figure 3b,d). In addition, we also tested the effects of binders on the uptake of NE by cancer cells as NE needs to enter the cytoplasm to stimulate cell proliferation. First, we tested the uptake ability of three different cancer cells on NE, the result showing that, compared with the A549 cell and SK-BR-3 cell, PC-3 showed the highest ability to uptake NE (data not shown). To test the block function of binders on NE uptake by cancer cells, we incubated IgG1 1C10 and V_H_-Fc 1D1.43 with NE at room temperature for 20 min, separately, then added the mixture to a PC-3 tumor cell followed with another 24 h incubation. The results showed that both IgG1 1C10 and V_H_-Fc 1D1.43 could significantly decrease the NE uptake by cancer cells (Figure 4) and the inhibition abilities were comparable. As NE plays a similar role in fibroblast proliferation, we tested whether NE binders could inhibit the fibroblast differentiation promoted by NE. The wound-healing assay showed that the NE treatment could facilitate the wound closure and the function of the NE was inhibited by IgG1 1C10 and V_H_-Fc 1D1.43, and the ratio of NE–Abs for inhibiting the differentiation of the fibroblast induced by NE was about 1:1.25 (Figure 5). These results suggested that IgG1 1C10 and V_H_-Fc 1D1.43 are potent antibody-based inhibitors of neutrophil elastase and are very promising in novel therapy for cancer and inflammatory-related diseases.

### 2.5. Epitope Mapping of IgG1 1C10 and V_H_-Fc 1D1.43

To test the binding epitope of IgG1 1C10 and V_H_-Fc 1D1.43, the competition Blitz and peptide array-based conformational epitope mapping was performed. We tested the competition effects of IgG1 1C10 and V_H_-Fc 1D1.43 for binding to human NE by BLItz (Figure 6a); it showed that IgG1 1C10 and V_H_-Fc 1D1.43 did not compete with each other, suggesting that the epitope of these two binders is different and this was confirmed by the conformational epitope mapping (Figure 6b and Appendix A). The conformational epitope mapping result showed that IgG1 1C10 had the antibody response against epitope-like spot patterns formed by adjacent lengths with the consensus motifs RPHAWPF, VRVVLGAHNLSRR, FENGYD, and VQVAQLPAQGRR at all peptide lengths. However, the interactions with peptides with the consensus motifs VRVVLGAHNLSRR, VQVAQLPAQGRR, and RRSNVCTLVRGR likely resulted from a non-specific ionic antibody binding due to shorter basic RR or RGR motifs in these consensus motifs. Additionally, motif RPHAWPF showed clearer spot morphologies than motif FENGYD (Appendix A). Therefore, we predicted that IgG1 1C10 bound to an epitope motif of ‘RPHAWPF’, which was located at the connecting hinge region of NE S1 and S2 subsites and was distal from the NE active site as defined by the catalytic triad (H57-D102-S195) (Figure 6c), indicating that the inhibition effects of IgG1 1C10 on NE may have been triggered by the NE conformational change upon its binding. By contrast, V_H_-Fc 1D1.43 showed a high antibody response to the ‘IFENGYD’ motif at all peptide lengths, which was adjacent to the active site (Figure 6d and Appendix A), indicating that the inhibition effect of V_H_-Fc 1D1.43 may have been due to directly blocking the accession of the substrate to the active site.

## 3. Discussion

NE was mostly reported to cause tissue damage and alter the extracellular matrix remodeling process in many lung-related inflammation diseases such as acute lung injury, pneumonia, and COPD. In addition, researchers have demonstrated that NE could induce fibroblast differentiation, which is correlated with lung fibrosis [41]. More recently, several reports showed that the high expression and activity of NE are related to cancer progressions such as Lung cancer, breast cancer, and prostate cancer. NE could directly promote cellular proliferation by targeting IRS-1, which activates the PI3K proliferation pathway both in fibroblasts and lung cancer cells [42]. Antibody-based therapy is a very efficient and promising treatment for cancer and, approximately, 30 therapeutic monoclonal antibodies have been approved by the FDA over recent decades. Moreover, there are many other ways that mAbs can be employed in cancer treatment such as targeting pro-tumorigenic molecules in the tumor microenvironment and antibody–drug conjugates (ADC).

In this study, we demonstrated and characterized two fully human anti-NE antibodies, V_H_ 1D1.43 and Fab 1C10, with a high-affinity of KD in the nanomolar range. These two antibodies exhibited good properties on aggregation resistance. V_H_ 1D1.43 showed a good developability as its stability at a high concentration after a long incubation at 37 °C without shaking (static standard incubation) and low aggregation. As for Fab 1C10, even though it showed an aggregation at 4 °C, such an aggregation disappeared after a 37 °C incubation at a high concentration. The reason for this reversible phenomenon may have been due to the temperature change during the purification process and this was recovered during the 37 °C incubation (Figure 1). Moreover, these two antibodies were able to specifically bind to human neutrophil elastase and did not bind to BSA, MPO, nor PR3 (Figure 2), indicating a low potential off-target toxicity for in vivo use. After converting to V_H_-Fc and IgG1, the avidities of these two antibody were elevated about 4-fold and 60-fold separately (Figure 1). The two antibodies were fully human; therefore, they likely were less immunogenic and could be much safer than chemical inhibitors.

The function of NE on cancer cell proliferation happens after it entry into the cancer cell cytoplasm and cleavage the IRS-1 to active the PI3K–AKT proliferation pathway. Therefore, the strategy of antibody therapy against NE is either by blocking the cancer cell uptake or inhibiting enzyme activity. We tested the inhibition of the two antibodies on enzyme activity; the V_H_-Fc 1D1.43 antibody showed a similar IC_50_ level compared with the positive inhibitor SPCK, and IgG1 1C10 showed a 2.6-fold higher level than a positive inhibitor. Both antibodies showed the block effect on the cancer cell uptake at the 1:3 NE–Abs ratio (Figure 4). The wound healing experiment also showed that these two antibodies could inhibit the differentiation of a fibroblast, which was promoted by NE at the 1:1.25 NE–Abs ratio (Figure 5). These results suggested that V_H_-Fc 1D1.43 and IgG1 1C10 could be promising antibodies for antibody therapy in cancer and inflammatory diseases.

We did not localize the antibodies epitope by the X-ray method, as the hNE–antibody complex was found hard to form the crystal. Instead, we conducted competition BLItz and conformational epitope mapping to predict the binding site of V_H_-Fc 1D1.43 and IgG1 1C10 with hNE. The predicted epitope motif of IgG1 1C10 was “RPHAWPF” and V_H_ 1D1.43 was “IFENGYD” (Figure 6). As the epitope of IgG1 1C10 was not near the enzyme active site, the possible mechanism of inhibiting the NE function by IgG1 1C10 was that IgG1 1C10 binding to hNE induced the change of NE conformation and caused it to lose its catalytic activity. On the contrary, the binding site of V_H_-Fc 1D1.43 was located at adjacent regions of the NE active site, so it could directly obstruct the access of the substrate and inhibit NE activity. By comparing the two antibodies functions, IgG1 1C10 was better than V_H_-Fc 1D1.43, which indicated that changing NE conformation may be better than directly blocking the active site by inhibiting the NE function.

In conclusion, we identified two fully human antibodies, the V_H_ domain and Fab fragment, that showed a high-affinity and specificity to human neutrophil elastase. The potent inhibition of NE enzyme activity and blocking effects on the NE uptake by cancer cell and fibroblast differentiation makes them promising immunotherapy candidates for cancer and inflammatory diseases.

## 4. Materials and Methods

### 4.1. Protein Expression and Purification (E. coli and Expi 293 Cell)

The ELANE gene was synthesized by IDT (Coralville, IA, USA), then cloned into pSecTag expression vector, which fused with human IgG1 Fc. For the conversion of IgG1 from Fab, the heavy chain and light chain of Fab were amplified and re-cloned into the pcDNA-IgG1 vector. For transient expression, the plasmid was transfected into Expi293 cells by PEI, and purified by Protein A resin (GenScript, Piscataway, NJ, USA). The expression and purification of V_H_ and Fab binders were performed in *E. coli* Top10F’ bacterial with 1 mM IPTG induction at 30 °C for 16 h. Bacterial pellets were lysed by Polymyxin B (Sigma-Aldrich, St. Louis, MO, USA) and the supernatant was loaded on Ni-NTA column (GE Healthcare, Chicago, IL, USA) for purification.

### 4.2. Panning and Screening from Two Large Phage Libraries

To select antibodies against neutrophil elastase, two large V_H_ and Fab phage libraries were used for panning separately. Briefly, 10 μg NE-Fc was coated in 96-well Protein G microplate in PBS at 4 °C overnight. Blocking coated the Protein G plate and phage library with 5% milk for 1 h at room temperature, then phage library was added into NE-Fc-coated Protein G plate and incubated at room temperature for 1 h, and then washed by PBST. After washing, phages were eluted by 0.1 M, PH3 Glycine, and neutralized with 1 M, PH8 Tris-HCl. For the second and third round panning, 5 μg and 2.5 μg of NE-Fc were used as antigen. After the third round panning, 192 individual clones were screened for binding with NE-His protein by soluble phage ELISA and 11 unique Fab and 4 V_H_ were identified.

### 4.3. Size-Exclusion Chromatography (SEC)

The purity and structure of the antibodies were analyzed by Superdex 200 Increase 10/300 GL chromatography (GE Healthcare, Chicago, IL, USA). The standard proteins used for calibration and their molecular weights were: Ferritin (M_r_ 440 kDa), Aldolase (M_r_ 158 kDa), Conalbumin (M_r_ 75 kDa), Ovalbumin (M_r_ 44 kDa), Carbonic anhydrase (M_r_ 29 kDa), and Ribonuclease A (M_r_ 13.7 kDa) at 3 mg/mL. A 500 μL sample mix containing above proteins was loaded to the column and separated by the ÄKTA explorer machine (GE Healthcare, Chicago, IL, USA). For the antibody analysis, 100 μL of filtered antibodies (2 mg/mL) in 1 × DPBS (Dulbecco’s phosphate-buffered saline, Gibco, Waltham, MA, USA) was analyzed. Antibodies were eluted by DPBS buffer at a flow rate of 0.5 mL/min.

### 4.4. Dynamic Light Scattering (DLS)

The aggregation resistance of V_H_ 1D1.43 and Fab 1C10 was measured by dynamic light scattering (DLS). The buffer was changed to DPBS and filtered by a 0.22 μm filter. The antibody concentration was adjusted to 1 mg/mL. In total, a 500 μL antibody sample was incubated at 37 °C without shaking. On day 0 and day 7, samples were taken for DLS measurements on Zetasizer Nano ZS ZEN3600 (Malvern Instruments Limited, Westborough, MA, USA) to determine the antibody size distribution.

### 4.5. Enzyme-Linked Immunosorbent Assay (ELISA)

NE protein was coated in 96-well microplates at 50 ng/well in PBS at 4 °C overnight and blocking with 5% milk in PBS for 2 h at room temperature. For the soluble Fab/V_H_ binding assay, the 3-fold serially diluted Fab/V_H_ were added and incubated for 1 h at 37 °C and further incubated with anti-FLAG M2-peroxidase (HRP) antibody (A8592, Sigma-Aldrich, St. Louis, MO, USA) for another 1 h. For the IgG1 binding assay, HRP-conjugated goat anti-human IgG1 Fc (Sigma-Aldrich, St. Louis, MO, USA) was used for detection. Finally, the reaction was developed by 3,3′,5,5′-tetramethylbenzidine (TMB, Sigma-Aldrich, St. Louis, MO, USA) and was stopped by TMB stop buffer (ScyTek Laboratories, Logan, UT, USA) and recording absorbance at 450 nm. The experiment was performed in duplicate.

### 4.6. BLItz

The affinity and avidity of the anti-hNE antibody were detected by biolayer interferometry BLItz (ForteBio, Menlo Park, CA, USA). Briefly, Dulbecco’s PBS (DPBS) was used to establish baseline for 30 s, and streptavidin biosensor (ForteBio) was coated with 16.7 ug/mL recombinant NE-Biotin for 2 min. Different doses of Fab and IgG1 were used for association and monitored for 2 min to measure the affinity and avidity. Antigen-coated biosensors with PBS were served as reference control. The dissociation was monitored in DPBS for 4 min.

### 4.7. Enzyme Activity Inhibition Assay

The inhibition effects of antibodies on enzyme activity were determined by an Elastase Inhibitor Screening Kit (Sigma-Aldrich). Briefly, antibodies were diluted to the desired concentration by assay buffer, then 50 μL of neutrophil elastase solution was added and incubated at 37 °C for 45 min. The SPCK NE inhibitor was used as positive control. After incubation, 25 μL of substrate solution was added to each reaction well and the fluorescence was measured in kinetic mode for 30 min. The result was calculated by %Relative inhibition.

### 4.8. Cells and Flow Cytometry

The human prostate adenocarcinoma cell line PC-3 was purchased from ATCC. Additionally, the cells were maintained in F-12k medium plus 10% FBS and 1% penicillin/streptomycin. Cells were seeded at a concentration of 1 × 10^5^ cells/well in a 24-well plate and allowed to adhere overnight and were then incubated in serum-free medium for 24 h. Neutrophil elastase was labeled with FITC by a FITC protein labeling kit (Thermo Fisher, Waltham, MA, USA) according to the manufacturer’s instruction. Cells were treated with 30 nM NE-FITC with or without 100 nM binders for 1 h; then, the cells were analyzed by flow cytometry (BD Bioscience, San Jose, CA, USA). The data were analyzed using FlowJo Software.

### 4.9. Wound Healing Assay

LL 47 fibroblast cells were seeded into 24-well plate and grown to confluence in complete F-12K medium. After being starved for 1 h in serum-free medium, the cells were wounded with 200 µL pipet tips, washed with PBS, and treated with NE with or without binders at different doses for 24 h. Images were captured under a 4 × objective microscope (Olympus microscope, Westborough, MA, USA). Wound areas were quantified by using Image J software.

### 4.10. Conformational Epitope Mappings

Conformational epitope mapping was proceeded by PEPperPRINT GmbH (Heidelberg, Germany). Briefly, the sequence of hNE was elongated with neutral GSGSGSG linkers at the C- and N-termini. The elongated antigen sequence was converted into 7, 10, and 13 amino acid peptides with peptide–peptide overlaps of 6, 9, and 12 amino acids. Human antibodies IgG1 1C10 and V_H_-Fc 1D1.43 at a concentration of 10 μg/mL and 30 μg/mL were incubated with antigen peptides for 16 h at 4 °C, followed by goat anti-human IgG DyLight680 incubation for 45 min at room temperature. The result was read by Innopsys InnoScan 710-IR Microarray Scanner.

### 4.11. Molecular Docking for Predicting Binding Models of Fab 1C10 and VH 1D1.43

Fab 1C10 and VH 1D1.43 structures were modeled in SWISS-MODEL [43,44], followed by energy minimization. Z-DOCK program was used for docking Fab 1C10 and VH 1D1.43 onto NE, whose structure was resolved as (PDB: 3Q76). The antibody binding and blocking regions on NE were based on the experimental epitope mapping results. Z-DOCK output the top 10 optimal poses, which were visually scrutinized for interaction interface compatibility and side-chain clashes, with selections of the most favorable pose as the binding models. The structural figures were prepared by PyMol 2.5.

### 4.12. Statistical Analysis

Statistical analyses, including EC_50_ and IC_50_, were performed by GraphPad Prism. Experiments were repeated a minimum of three times. Differences were considered statistically significant when *p* < 0.05.

## Figures and Tables

**Figure 1 ijms-22-11136-f001:**
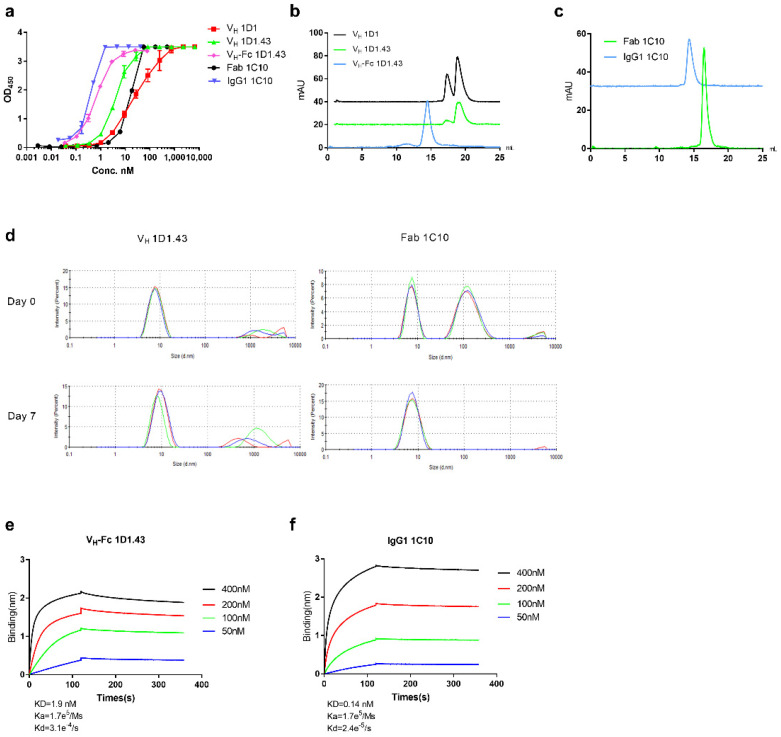
Binding and aggregation characterization of different format anti-hNE antibodies. (**a**) V_H_ 1D1.43, Fab 1C10, V_H_-Fc 1D1.43, and IgG1 1C10 binding to recombinant NE measured by ELISA. Experiments were performed in duplicate. (**b**) Aggregation evaluation of V_H_ 1D1, V_H_ 1D1.43, and V_H_-Fc 1D1.43 measured by SEC. (**c**) Aggregation evaluation of Fab 1C10 and IgG1 measured by SEC. (**d**) Aggregation evaluation of V_H_ and Fab measured by DLS. Antibodies were evaluated at a concentration of 1 mg/mL. (**e**,**f**) Kinetics of V_H_-Fc 1D1.43 (**e**) and IgG1 1C10 (**f**) binding to recombinant NE measured by BLItz. Values were reported as the mean ± SD.

**Figure 2 ijms-22-11136-f002:**
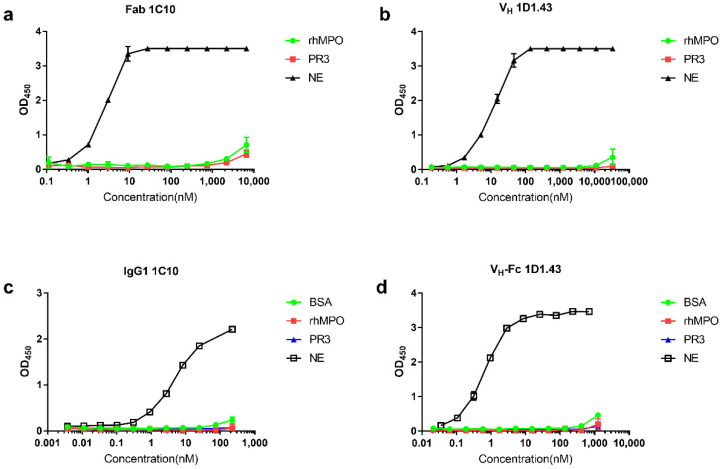
Specificity of V_H_/V_H_-Fc and Fab/IgG1 with recombinant human NE. (**a**,**b**) Binding of Fab 1C10 (**a**) and V_H_ 1D1.43 (**b**) to rhNE, rhMPO, and PR3 measured by ELISA. (**c**,**d**) Binding of IgG1 1C10 (**c**) and V_H_-Fc 1D1.43 (**d**) to rhNE, rhMPO, BSA, and PR3 measured by ELISA. rhNE, rhMPO, BSA, and PR3 were coated in 96-well microplates at 50 ng/well. Experiments were performed in duplicate. Values were reported as the mean ± SD.

**Figure 3 ijms-22-11136-f003:**
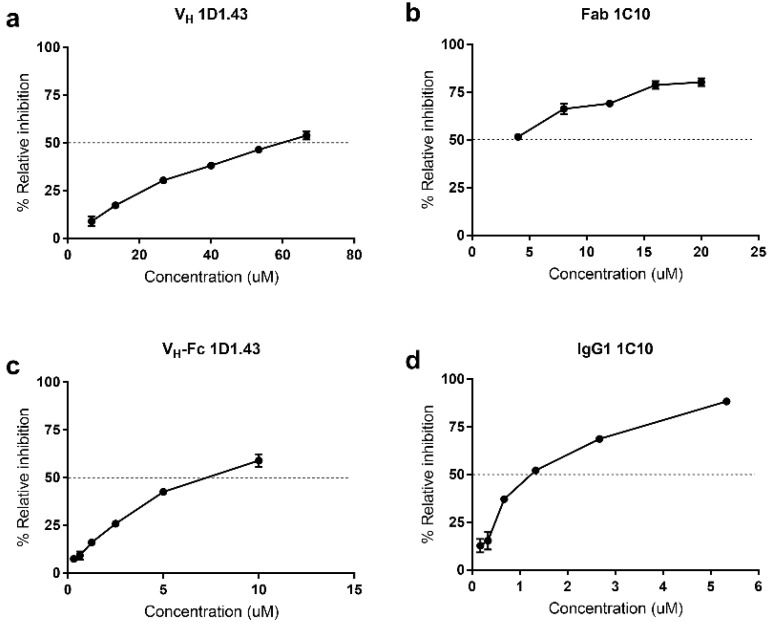
Inhibition effects of V_H_/V_H_-Fc 1D1.43 and Fab/IgG1 1C10 on enzyme activity. (**a**,**c**) Inhibition of V_H_ 1D1.43 (a) and V_H_-Fc 1D1.43 (**c**) on NE enzyme activity. (**b**,**d**) Inhibition of Fab 1C10 (**b**) and IgG1 1C10 (**d**) on NE enzyme activity. Each experiment was performed in duplicate. Calculation: choose two time points (T1 and T2) in the linear range, determine the Fluorescence (FLU) at each time (FLU1 and FLU2) and use them to determine the Slope. Slope = (FLU2 − FLU1)/(T2 − T1) = ΔFLU/minute. %Relative inhibition = (Slope_EC_ − Slope_SM_)/Slope_EC_ × 100%; Slope_EC_ = the slope of the Sample inhibitor and Slope_SM_ = the slope of the Enzyme control. Values were reported as the mean ± SD.

**Figure 4 ijms-22-11136-f004:**
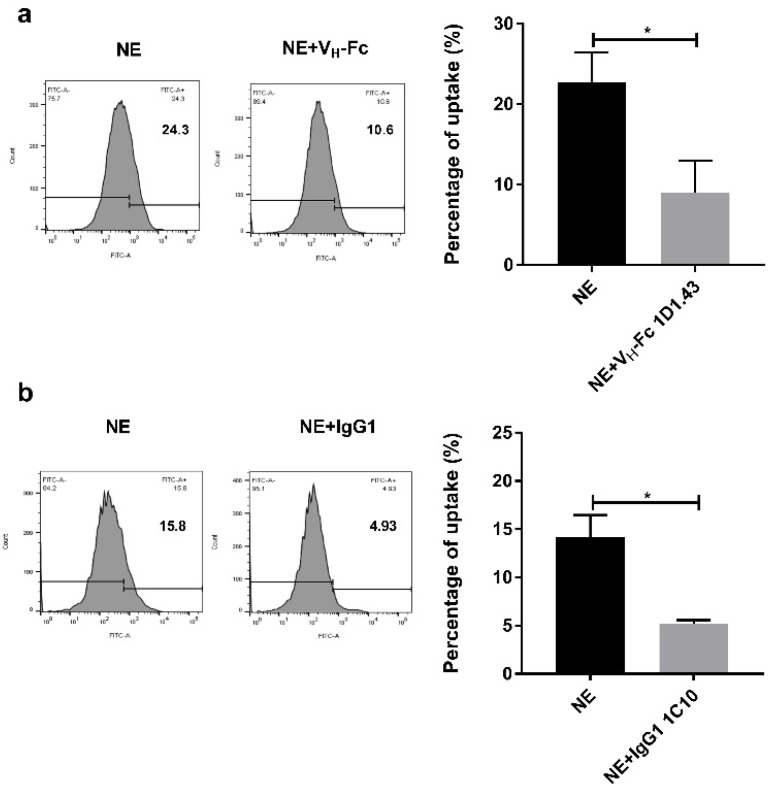
V_H_-Fc 1D1.43 and IgG1 1C10 inhibited NE uptaking by tumor cell. (**a**,**b**) Inhibition of VH-Fc 1D1.43 (**a**) and IgG1 1C10 (**b**) on NE uptaking by PC-3 tumor cell measured by flow cytometry. PC-3 cells were plated in a 24-well plate and incubated with serum-free medium for 24 h, then treated with 30 nM NE-FITC with or without 100 nM antibody proteins for 1 h. Experiments were performed in duplicate; unpaired Student’s *t*-test, * *p* < 0.05. Values were reported as the mean ± SD.

**Figure 5 ijms-22-11136-f005:**
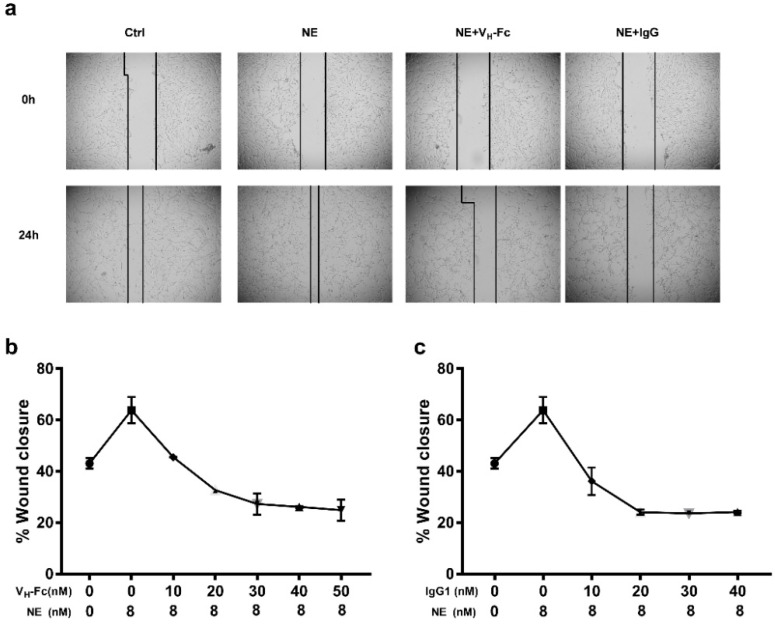
V_H_-Fc 1D1.43 and IgG1 1C10 inhibited NE-promoted fibroblast differentiation. (**a**) Representative pictures of wound-healing assay. LL47 fibroblasts were plated into a 24-well plate and scratched, treated with 8 nM NE with or without antibody proteins at different concentrations for 24 h. (**b**,**c**) V_H_-Fc 1D1.43 (**b**) and IgG1 1C10 (**c**) inhibited NE-promoted LL 47 fibroblast differentiation. Experiments were performed in duplicate. Values were reported as the mean of percent wound closure ± SD.

**Figure 6 ijms-22-11136-f006:**
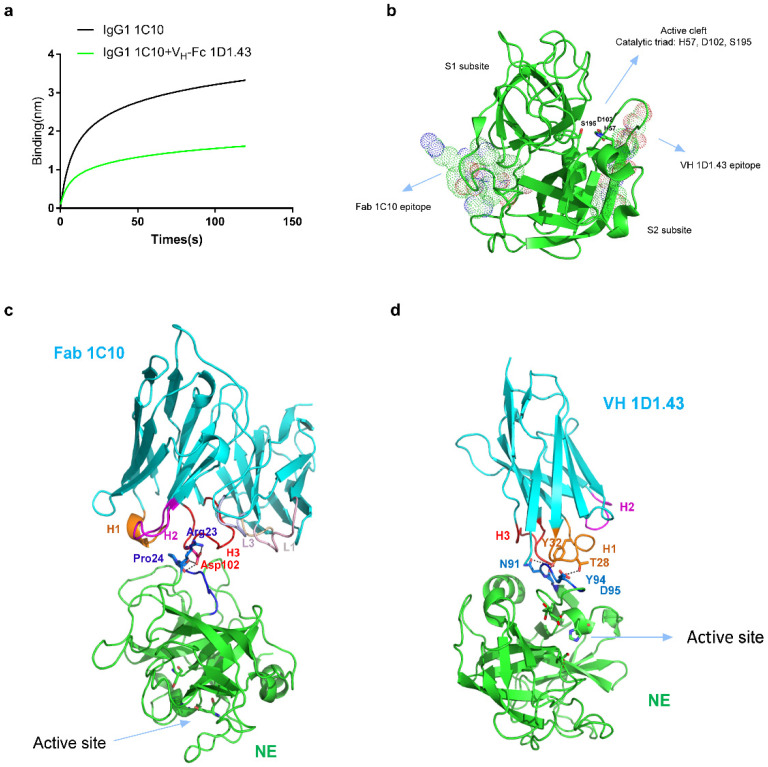
Epitope mapping of V_H_-Fc 1D1.43 and IgG1 1C10 by using conformational epitope mappings. (**a**) Competition between V_H_-Fc 1D1.43 and IgG1 1C10 measured by BLItz. (**b**) The binding region of V_H_ 1D1.43 and Fab 1C10 on hNE (PDB: 3Q76) based on the conformational epitope mapping results with the binding motifs highlighted by dot presentations. The enzymatic sites are shown by the stick models. (**c**,**d**) The detailed binding model of Fab 1C10 and V_H_ 1D1.43 on hNE, as predicted by Z-DOCK programs. Human NE is represented as green cartoons with active site highlighted by sticks and binding epitope highlighted by blue color. Fab 1C10 and V_H_ 1D1.43 are represented as cyan cartoons with highlighted colored CDR loops.

**Table 1 ijms-22-11136-t001:** Blitz results of human neutrophil elastase antibodies.

Antibody	k_on_ (M^−1^ s^−1^) ^1^	k_off_ (s^−1^) ^1^	KD (nM) ^1^
V_H_ 1D1	5.9 × 10^4^	4.1 × 10^−3^	69.7
V_H_ 1D1.43	7.2 × 10^4^	6 × 10^−4^	8.4
V_H_-Fc 1D1.43	1.7 × 10^5^	3.1 × 10^−4^	1.9
Fab 1C10	4.9 × 10^4^	4.1 × 10^−4^	8.5
IgG1 1C10	1.7 × 10^5^	2.4 × 10^−5^	0.14

^1^ Mean kinetic rate constants (k_on_, k_off_) and equilibrium dissociation constants (KD = k_off_/k_on_) were determined from curve fitting analyses of BLItz results.

## Data Availability

The data presented in this study are available upon request from the corresponding author.

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
