# Peer review of "Human Antibody Domains and Fragments Targeting Neutrophil Elastase as Candidate Therapeutics for Cancer and Inflammation-Related Diseases"

_ijms, 2021, doi:10.3390/ijms222011136_

Round 1

Reviewer 1 Report

Dimitrov and coworkers generated human antibody domains and fragments targeting neutrophil  elastase as candidate therapeutics for cancer and inflammation related diseases. While neutrophil elastase is known to play multiple positive roles in the human body, its overactivity is connected to many serious diseases. Therefore, neutrophil elastase inhibition as a therapeutic strategy widely investigated for the treatment/prevention of immune disorders, cancers, chronic kidney disease, respiratory disorders, etc. The authors identified two fully human antibodies VH domain and Fab fragment that shows high affinity and specificity to human neutrophil elastase. The authors also identified the potential mechanism by which these antibodies inhibit neutrophil elastase. The discovered antibodies Fab and VH  also fused to human IgG1 Fc to improve their avidity and in vivo half-life.

Introduction- there are many neutrophil elastase inhibitors are currently in various pre-clinical and clinical stages. Providing more example of inhibitors specially the ones are in clinical phases provides readers with better picture of research in this area.

page 9 line 228- please  add the rpm used for incubation

page 10 line 287 ( 4.3-SEC) please add more information to this section such as,  gradient and buffer used.

page 10 line 294 (4.4-DLS) please add more information to this section such rpm used for incubation and the buffer used for DLS analysis.

 Overall, the paper is well written and contains a wealth of data. Hence the manuscript can be recommended for publication.

Author Response

Response to Reviewer 1 Comments:

Comments and suggestions for Authors:

Comments:

Dimitrov and coworkers generated human antibody domains and fragments targeting neutrophil elastase as candidate therapeutics for cancer and inflammation related diseases. While neutrophil elastase is known to play multiple positive roles in the human body, its overactivity is connected to many serious diseases. Therefore, neutrophil elastase inhibition as a therapeutic strategy widely investigated for the treatment/prevention of immune disorders, cancers, chronic kidney disease, respiratory disorders, etc. The authors identified two fully human antibodies VH domain and Fab fragment that shows high affinity and specificity to human neutrophil elastase. The authors also identified the potential mechanism by which these antibodies inhibit neutrophil elastase. The discovered antibodies Fab and VH also fused to human IgG1 Fc to improve their avidity and in vivo half-life.

We appreciate the reviewer’s comments and suggestions. Changes in the text were marked by using the “Track Changes” function.

Point 1: Introduction- there are many neutrophil elastase inhibitors are currently in various pre-clinical and clinical stages. Providing more example of inhibitors specially the ones are in clinical phases provides readers with better picture of research in this area.

Response 1: We agree with the reviewer’s comments and the following paragraph was added to the introduction on page 2 lines 61-71.

“In recent years, many novel orally available neutrophil elastase inhibitors were designed and some already entered phase II clinical trials for a variety of pulmonary diseases. MPH-966 significantly suppressed NE activity and reduced pro-inflammatory cytokines in a 5-FU-induced intestinal mucositis mouse model [28]. In a preclinical study, BAY-85-8501 proved to be able to prevent the development of lung injury and inflammation induced by NE in acute lung injury [29]. Phase I clinical study also showed safety and tolerability profile in patients with non-cystic fibrosis bronchiectasis [30]. POL6014, a novel NE inhibitor developed for patients with cystic fibrosis showed a safety and tolerability profile and a significant reduction of active NE after single dosing in phase I clinical trial [31]. However, all these studies were designed for pulmonary diseases. We are not aware of any cancer related studies.”

Point 2: page 9 line 228- please add the rpm used for incubation

Response 2: The incubation condition is 37°C without shaking (static standard incubation), we added the description accordingly on page 10 line 252 in the revised manuscript.

Point 3: page 10 line 287 (4.3-SEC) please add more information to this section such as, gradient and buffer used.

Response 3: The detailed information was added on page 11 line 312 in the revised manuscript.

Previous description:

“4.3. Size-exclusion chromatography (SEC)

The purity and structure of antibodies were analyzed by Superdex 200 Increase 10/300 GL chromatography (GE Healthcare). The protein molecular standardization of the column was measured by Ferritin (Mr 440 kDa), Aldolase(Mr 158 kDa), Conalbumin (Mr 75 kDa), Ovalbumin (Mr 44 kDa), Carbonic anhydrase (Mr 29 kDa), Ribonuclease A (Mr 13.7 kDa). 200 ug filtered proteins in DPBS were analyzed at a flow rate of 0.5 ml/min.”

Revised paragraph:

“4.3. Size-exclusion chromatography (SEC)

The purity and structure of the antibodies were analyzed by Superdex 200 Increase 10/300 GL chromatography (GE Healthcare). The standard proteins used for calibration and their molecular weights are:  Ferritin (Mr 440 kDa), Aldolase (Mr 158 kDa), Conalbumin (Mr 75 kDa), Ovalbumin (Mr 44 kDa), Carbonic anhydrase (Mr 29 kDa), Ribonuclease A (Mr 13.7 kDa) at 3mg/ml. A 500 μl sample mix containing above proteins were loaded to the column and separated by the ÄKTA explorer machine (GE Healthcare). For the antibody analysis, 100 μl filtered antibodies (2mg/ml) in 1×DPBS (Dulbecco's phosphate-buffered saline, Gibco) were analyzed. Antibody was eluted by DPBS buffer at a flow rate of 0.5 ml/min.”

Point 4: page 10 line 294 (4.4-DLS) please add more information to this section such rpm used for incubation and the buffer used for DLS analysis.

Response 4: For the DLS experiment, the incubation condition of antibody is at 37°C without shaking and the buffer is DPBS. We added the description accordingly on page 11 line 321 in the revised manuscript.

Previous description:

4.4. Dynamic light scattering (DLS)

The aggregation resistance of VH 1D1.43 and Fab 1C10 was detected by DLS. The antibody at a concentration of 1 mg/ml was incubated at 37°C, detect the size distribution of antibodies on different incubation time points by Zetasizer Nano ZS ZEN3600 (Malvern Instruments Limited, Westborough, MA).”

Revised paragraph:

“4.4. Dynamic light scattering (DLS)

The aggregation resistance of VH 1D1.43 and Fab 1C10 was measured by dynamic light scattering (DLS). The buffer was changed to DPBS and filtered by a 0.22 μm filter. The antibody concentration was adjusted to 1 mg/ml. 500 μl antibody sample was incubated at 37°C without shaking. On day 0 and day 7, samples were taken for DLS measurements on Zetasizer Nano ZS ZEN3600 (Malvern Instruments Limited, Westborough, MA) to determine the antibody size distribution.

Overall, the paper is well written and contains a wealth of data. Hence the manuscript can be recommended for publication.

Reviewer 2 Report

Chu et al. isolated two human anti-neutrophil elastase (NE) antibodies with high affinity of KD in the nanomolar range, which inhibits the enzymatic activity of NE. Both of the antibodies exhibited blockage of intracellular uptake of NE on cancer cells, which is a preceding process for cancer proliferation. In addition, these antibodies also displayed inhibition of fibroblast differentiation.

As a reviewer, I would suggest that the authors be clear on the following issues to be acceptable for the IJMS:

  1. As cancer therapeutic candidates, against which NE in our body the two antibodies target i.e. NE on neutrophils or NE on cancer cells? (Figure 4)? And how widely NE is distributed in either normal person or cancer patients? This issue may be resolved in the introduction section if the antibodies are to be developed as a cancer immunotherapy (lines 58-68).
  2. As therapeutic candidates for inflammatory diseases against which NE in our body the two antibodies target (Figure 5)? And how widely NE is distributed in either normal person or patients with inflammation?

Minor corrections:

Line 95: After three rounds panning --> After three rounds of panning

Line 139: The estimate IC50 --> The estimated IC50

Line 226: with high affinity at nM level --> with high affinity of KD in the nanomolar range

Line 243: Both of antibodies --> Both antibodies

Line 267: Ecoli --> E. coli

Author Response

Response to Reviewer 2 Comments:

Comments and suggestions for Authors:

Comments:

Chu et al. isolated two human anti-neutrophil elastase (NE) antibodies with high affinity of KD in the nanomolar range, which inhibits the enzymatic activity of NE. Both of the antibodies exhibited blockage of intracellular uptake of NE on cancer cells, which is a preceding process for cancer proliferation. In addition, these antibodies also displayed inhibition of fibroblast differentiation.

We appreciate the reviewer’s comments and suggestions. Changes in the text were marked by using the “Track Changes” function.

As a reviewer, I would suggest that the authors be clear on the following issues to be acceptable for the IJMS:

Point 1: As cancer therapeutic candidates, against which NE in our body the two antibodies target i.e. NE on neutrophils or NE on cancer cells? (Figure 4)? And how widely NE is distributed in either normal person or cancer patients? This issue may be resolved in the introduction section if the antibodies are to be developed as a cancer immunotherapy (lines 58-68).

Response 1: We thank the reviewer for his useful comments. Neutrophils are the dominant cellular source of NE. The two antibodies target NE secreted by neutrophils. We added the following description into the introduction section accordingly to describe the NE level in cancer patients and patients with inflammatory diseases such as COPD with comparisons to healthy patients on page 2 line 72-81 in the revised manuscript.

“In many cancer types, both expression and activity of neutrophil elastase are upregulated. The number of neutrophils and neutrophil to lymphocyte (N/L) ratio in the peripheral blood is significantly higher in patients with lung cancer with or without COPD than in patients with COPD or healthy individuals. However, the cell count of neutrophils in BAL fluid is significantly lower in patients with lung cancer than in individuals with COPD. Notably, the NE level in serum and BAL fluid is five and threefold, respectively, greater in lung cancer patients compared to patients with COPD [14]. A strong NE proteolytic fingerprint was also found in the colon adenocarcinoma proteome [32], and elevated NE has a positive correlation with the poor response to trastuzumab therapy [33]. In summary, cancer is a potent inducer of NE.”

Point 2: As therapeutic candidates for inflammatory diseases against which NE in our body the two antibodies target (Figure 5)? And how widely NE is distributed in either normal person or patients with inflammation?

Response 2: We thank the reviewer for his useful comments. The two antibodies target the NE secreted by neutrophils as therapeutic candidates for inflammatory disease. NE has a negative function both in cancers and inflammatory diseases. We described NE function in cancer and inflammatory disease on Page1-2 lines 38-48. The following description of NE level and distribution in patients compared with healthy person was added to the revised manuscript on page 1-2 Lines 42-45.

“The level of NE activity and the cell count of neutrophils were highly elevated in sputum plasma from cystic fibrosis patients compared with healthy individuals [12, 13]. Similarly, COPD patients also have a two-fold higher level of NE activity than healthy individuals in serum and bronchoalveolar lavage (BAL) fluid [14].”

Minor corrections:

Point 3: Line 95: After three rounds panning --> After three rounds of panning

Response 3: Thank you for your comment, the manuscript has been revised accordingly in the revised manuscript on page 3 line 118.

Point 4: Line 139: The estimate IC50 --> The estimated IC50

Response 4: Authors thanks for your comments, we have correct the manuscript accordingly in the revised manuscript on page 5 line 162 and 164.

Point 5: Line 226: with high affinity at nM level --> with high affinity of KD in the nanomolar range

Response 5: We have revised the manuscript accordingly in the revised manuscript on page 9 line 249.

Point 6: Line 243: Both of antibodies --> Both antibodies

Response 6: We correct it accordingly in the revised manuscript on page 10 line 267

Point 7: Line 267: Ecoli --> E. coli

Response 7: We have revised it accordingly in the revised manuscript on page 10 line 291 and 297.